# Mutation Analysis of SARS-CoV-2 Variants Isolated from Symptomatic Cases from Andhra Pradesh, India

**DOI:** 10.3390/v15081656

**Published:** 2023-07-29

**Authors:** Mudhigeti Nagaraja, Kodavala Sireesha, Anagoni Srikar, Katari Sudheer Kumar, Alladi Mohan, Bhuma Vengamma, Chejarla Tirumala, Anju Verma, Usha Kalawat

**Affiliations:** 1State-Level VRDL, Department of Clinical Virology, Sri Venkateswara Institute of Medical Sciences, Tirupati 517 507, Andhra Pradesh, India; 2Regional Center for ISCP-NCDC, Department of Clinical Virology, Sri Venkateswara Institute of Medical Sciences, Tirupati 517 507, Andhra Pradesh, India; 3Department of Medicine, Sri Venkateswara Institute of Medical Sciences, Tirupati 517 507, Andhra Pradesh, India; 4Sri Venkateswara Institute of Medical Sciences, Tirupati 517 507, Andhra Pradesh, India; 5Department of Tuberculosis and Respiratory Diseases, Sri Balaji Medical College Hospital and Research Institute, Renigunta, Tirupati 517 507, Andhra Pradesh, India; 6Department of Clinical Virology, Sri Venkateswara Institute of Medical Sciences, Tirupati 517 507, Andhra Pradesh, India

**Keywords:** SARS-CoV-2, amino acid substitutions, mutations, Andhra Pradesh, India, circulating clades, S-gene sequencing

## Abstract

There has been a continuous evolution in the SARS-CoV-2 genome; therefore, it is necessary to monitor the shifts in the SARS-CoV-2 variants. This study aimed to detect various SARS-CoV-2 variants circulating in the state of Andhra Pradesh, India. The study attempted to sequence the complete S-gene of SARS-CoV-2 of 104 clinical samples using Sanger’s method to analyze and compare the mutations with the global prevalence. The method standardized in this study was able to amplify the complete length of the S-gene (3822 bp). The resulting nucleotide and amino acid mutations were analyzed and compared with the local and global SARS-CoV-2 databases using Nextclade and GISAID tools. The Delta variant was the most common variant reported in the present study, followed by the Omicron variant. A variant name was not assigned to thirteen samples using the Nextclade tool. There were sixty-nine types of amino acid substitutions reported (excluding private mutations) throughout the spike gene. The T95I mutation was observed predominantly in Delta variants (15/38), followed by Kappa (3/8) and Omicron (1/31). Nearly all Alpha and Omicron lineages had the N501Y substitution; Q493R was observed only in the Omicron lineage; and other mutations (L445, F486, and S494) were not observed in the present study. Most of these mutations found in the Omicron variant are located near the furin cleavage site, which may play a role in the virulence, pathogenicity, and transmission of the virus. Phylogenetic analysis showed that the 104 complete CDS of SARS-CoV-2 belonged to different phylogenetic clades like 20A, 20B, 20I (Alpha), 21A (Delta), 21B (Kappa), 21I (Delta), 21J (Delta), and 21L (Omicron).

## 1. Introduction

Severe acute respiratory syndrome coronavirus 2 (SARS-CoV-2) shares genetic similarity with bat and pangolin coronaviruses [1]. The present SARS-CoV-2 is more closely related to two other recent human pathogens; SARS-CoV and MERS-CoV, and shares 79% and 50% genetic similarity, respectively [2]. There has been a continuous evolution in the SARS-CoV-2 genome since it was first reported, particularly in the S-gene. It is necessary to monitor the shifts in the SARS-CoV-2 variants as they may affect virus transmissibility, virulence, immune protection, vaccine efficacy, etc. [3]. Because of the continuous accumulation of genetic mutations in the SARS-CoV-2 nucleotide sequence from time to time, different groups adopted different naming systems such as variants, lineages, clades, etc. For the convenience of non-scientific communities to monitor and implement appropriate preventive measures, one or more lineages with similar features are further designated as a “variant being monitored” (VBM), a “variant of high consequence” (VOHC), a “variant of concern” (VOC), or a “variant of interest” (VOI) [4]. For instance, “Omicron” (WHO nomenclature) is the currently circulating VOC named B.1.1.529 by Pango, GR/484A by GISAID, or 21K/21L/21M/22A/22B/22C/22D by Nextstrain classification systems [5]. This study aimed to detect various SARS-CoV-2 variants circulating in Andhra Pradesh and understand the changing trend of virus host interactions in terms of bonding and non-bonding interactions.

## 2. Materials and Methods

### 2.1. Sampling and S-Gene Amplification

A total of 1.8 million respiratory samples received from March 2020 to April 2022 were subjected to SARS-CoV-2 real-time reverse transcriptase polymerase chain reaction (rRT-PCR). As per the directions of the Indian Council of Medical Research (ICMR), two percent of positive and negative samples were stored at −80 °C. After obtaining the institute’s ethics committee clearance (IEC No. 1268, dt: 11 April 2022), a total of 900 samples were retrieved from the archive. All samples were re-tested (TaqPathTM COVID-19 Combo Kit, Thermo Fisher Scientific Inc., Waltham, MA, USA) to check for RNA integrity. A total of 502 samples that yielded a cycle threshold (Ct) value below 25 were subjected to S-gene-endpoint PCRs, standardized and validated in-house. Out of 502 samples, only 104 showed bright and sharp bands for all six targets by the end-point PCR and were subjected to sequencing.

To scaffold the complete S-gene (3822 bases), 14 primer sets (nCoV-2019_71_LEFT to nCoV-2019_84_RIGHT) were selected from the ARTIC Network. While standardizing the endpoint PCR and sanger sequencing protocols, there were gaps in the nucleotide sequence. To cover the entire sequence of the S-gene and reduce cost per sample, the number of PCR reactions was reduced to six, namely T1, T2, T3, T4, T5, and T6. The details of primers and the size and location of targets are given in Table 1.

Primers were synthesized at Synbio Technologies (Monmouth Junction, NJ, USA). All endpoint PCR reactions were performed in a 25 µL reaction volume containing TaqPath 4× PCR buffer (6.25 µL), forward and reverse primers (each 10 pmol/reaction), nuclease-free water (11.75 µL), and RNA template (5.0 µL). The PCR reaction was carried out in a PCR thermal cycler (Eppendorf, Hamburg, Germany) using the following thermal profiles: reverse transcription at 53 °C for 10 min, initial denaturation at 95 °C for 2 min, 45 cycles of denaturation (95 °C for 15 s), annealing (64 °C for 30 s), extension (72 °C for 30 s), and final extension at 72 °C for 10 min. PCR products were visualized by 1.5% agarose gel electrophoresis. 

### 2.2. Sequencing Procedure

PCR products were purified using ExoSAP-ITTM (Applied Biosystems, Waltham, MA, USA). Briefly, 2 µL of ExoSAP-ITTM was added directly to the 5 µL of each amplicon in a separate tube and incubated for 15 min at 37 °C, followed by 15 min at 80 °C. Two microliters of the cleanup product were used for cycling sequencing with BigDyeTM Terminator cycle sequencing kit 3.1 (Applied Biosystems, Waltham, MA, USA). The cycling sequencing was performed in a 0.2 mL PCR tube containing BigDyeTM Terminator (1.0 µL), respective sequencing primer (1.6 µL), 5× sequencing buffer (3.5 µL), respective ‘Exosap’ cleanup product (2.0 µL), and nuclease-free water (11.9 µL). The reaction was carried out with the following thermal profile conditions; initial denaturation (1 min), denaturation (10 s), annealing (5 s), extension (4 min) for 25 cycles, and final extension (10 min).

After the cycle sequencing reaction, the entire (20 µL) contents of the tube were transferred into a 1.5 mL centrifuge tube, 5 µL of 125 mM EDTA, and 60 µL of 100% ethanol were added, and the mixture was incubated for 15 min at room temperature. Centrifuge at 1870× *g* for 45 min at 4 °C, decant the contents, and air dry. Overall, 60 µL of 70% ethanol was added, centrifuged at 1870× *g* for 15 min at 4 °C, decanted, and air dried. We resuspended the samples in 20 µL formamide thoroughly for 15 s and loaded them on a 96-well ABI-sequencing barcode plate. The resuspended samples were denatured in a thermocycler for 2 min at 95 °C and 4 °C for 2 min. Capillary gel electrophoresis was performed for the separation and detection of the sequence of nucleotides on the 3500 genetic analyzers (Applied Biosystems, Waltham, MA, USA) using the standard sequencing method and POP7 polymer.

### 2.3. Sequence Analysis

A total of six fragments (T1 to T6) were obtained for each sample (n = 104). Base calling was performed with the data collection software (Applied Biosystems), and a threshold of 25% was considered for mixed base calls. All reactions with bad quality or short sequences were repeated from the endpoint PCR step. The reference Wuhan-1 genome was used to align with the resulting contigs (NC_045512.2). Applied Biosystems Sequencing Analysis Software (SeqScape4) was used for the analysis of both aligned and consensus sequences obtained after capillary gel electrophoresis. Aligned and consensus sequences (six fragments and one complete sequence) of each sample were exported and saved in FASTA format. A nucleotide similarity search was performed using the BLASTn (Basic Local Alignment Search Tool; nucleotide) tool in the NCBI online portal. The ends of the first (T1) and last (T6) PCR products of each sample contained additional sequences from ORF1ab and ORF3a sequences, respectively. Ugene software [6] was used to trim these sequences after they were aligned with the Wuhan-1 reference genome and submitted to NCBI GenBank. 

### 2.4. Phylogeny Tree Construction

The 104 complete sequences of SARS-CoV-2 were used for the construction of a phylogenetic tree using the Nextclade tool (version 2.9.1) [7]. This tool identifies differences between the query sequence and a reference sequence (Wuhan-Hu-1/2019 (MN908947)). Based on the changes in the nucleotide sequence, the software assigned a clade, lineage, or WHO name to each sample sequence. The software also reported any private or recombinant mutations (no name assigned) and potential sequence quality issues in the data. A rectangular phylogeny tree was constructed for all 104 complete S-gene sequences based on the WHO name, clade, and PANGO lineage naming systems.

### 2.5. Computational Biology Workup

Sixty-six unique SARS-CoV-2 spike gene sample variants were translated into protein variants, and these structures were modeled to build the structures by homology modeling through Modeller v10.4 [8]. Ten models of each of the unique 66 SARS-CoV-2 trimer variants were built by aligning the structural coordinates of the template structure with CLUSTALX [9]. Among the built structures, the best DOPE (Discrete Optimized Protein Energy) score with the least possible energy state was subjected to validation studies by PROCHECK [10] (Ramachandran plot analysis) (Appendix A), ProSA (Protein Structural Analysis), and ProQ (Protein Quality) analysis. Human Angiotensin Converting Enzyme 2 (hACE2) structure was retrieved from the Protein Data Bank (PDB) [11] and it was subjected to protein–protein docking with the best validated and modeled 66 trimer SARS-CoV-2 spike variants through ClusPro [12]. A total of 660 complexes were obtained, the best pose among each of the ten complexes of SARS-CoV-2 trimer spike with hACE2 and the unique mutation interactions or their influence were deciphered among the Alpha, Delta, Kappa, and Omicron variants. From each variant group (Alpha, Delta, Kappa, Omicron, and Other), one structure showing maximum genetic mutations was selected and colored, marked, and labeled using PyMOL in a Windows environment [13]. (Appendix A)

### 2.6. Statistical Analysis

Microsoft Excel Worksheet 2019 (Microsoft Office Suite for Windows) was used for data compilation, sorting, and outlier detection. An online scientific statistical program (Graph Pad Prism 9.4.0) was used to analyze and generate heatmaps, plots, and interaction patterns within the spike trimer and the spike-ACE2 proteins using one-way ANOVA.

## 3. Results and Discussion

Our center was the first COVID-19 testing lab identified in March 2020 in the state of Andhra Pradesh, India. In March and April, the daily testing limit was less than 500 samples, and subsequently, the capacity was increased to 10,000 samples per day. As per the laboratory data (Figure 1), the first COVID-19 wave started in April 2020, reached its peak in July–August 2020, and declined by the end of November 2020. The second wave started in March 2021 and declined by July 2021, whereas the third wave was observed from December 2021 to March 2022. The fourth peak was very small, with a daily sample load of less than 50. The first wave peak was broad (eight months), and the maximum daily positive percentage was below 28%; the second wave was shorter than the first wave, but the daily maximum positive percentage was up to 38%; whereas the third wave was shorter with a high daily positive percentage (56%) during the months of January and February. A possible reason for the broader first wave could be a lack of immunity despite strict lockdown measures imposed by the government. By the time the second wave hit, SARS-CoV-2 vaccination had started for healthcare workers and elderly populations, a few lockdown measures were relaxed, and some proportion of the population had immunity due to previous exposure. During the third wave, the lockdown was completely reversed, and the fear of the disease almost waned; hardly any disease preventive measures were followed by the public, and the majority of active populations had the vaccination. Since March 2022, the daily sample load for SARS-CoV-2 rRT-PCR testing has been less than 50 suspected cases per day, and the positivity has been close to zero on most days.

A total of 104 samples were processed for spike gene sequencing. The method we developed successfully amplified the complete length of the S-gene (3822 bp). In addition to this, there were around 350 nucleotide overlaps between the targets and at the ends of the spike gene, which helped to avoid the use of bidirectional sequencing. This additional sequence was trimmed during the post-sequence analysis steps using the Wuhan-1 genome (NC_045512.2) as a reference sequence. Trimmed sequences of complete and partial S-genes were submitted to NCBI GenBank. Total submissions of complete and partial S-gene sequences were 104 and 838, respectively. Accession numbers for the complete S-gene were ON644350–ON644384, ON651690–ON651724, and ON668127–ON668160. Accession numbers for partial S-genes were ON644612-ON644868, ON651730-ON652014, ON652017-ON652303, and ON680851-ON680920.

The obtained FASTA sequences were analyzed using the Nextclade database [7] to assign the clade, perform mutation calling, and perform sequence quality checks. The Spike protein from clinical samples displayed various changes such as frameshift mutations (1483; 2–29/sample), nucleotide deletions (717; 0–15), nucleotide insertions (6; 0–6), amino acid substitutions (1474; 1–29), amino acid deletions (239; 0–5), and amino acid insertions (1) (Figure 2). In the present study, it was observed that nucleotide changes accumulated as time progressed. Early in November 2020, there were only 14 changes, whereas by the end of March 2022, close to 40 nucleotide changes were observed as compared to the reference genome. It was observed that the SARS-CoV-2 was evolving at a rate of approximately two mutations per month in the global population [14,15,16]. Consistent with the above findings, the present study observed two nucleotide changes per month (November 2020 (14) to December 2021 (38); 24/12 = 2).

Out of 104 samples sequenced, a total of 14, 38, 8, 31, and 13 were identified as Alpha, Delta, Kappa, Omicron, and other lineages, respectively (Figure 3). The Kappa variant was reported only during the months of March and April months (2021). Alpha variants were detected in March, April, and May 2021, and the Delta variant was reported throughout the year 2021. Omicron, the second predominant variant, started at the end of December 2021, replaced other variants, and continued until April 2022 (the end of the study) (Figure 4). A variant name was not assigned to 13 samples using the Nextclade tool; however, the NCBI blast search revealed several matches (99 to 100%) reported from India, Hong Kong, the United States of America (USA), France, New Zealand, Kenya, Switzerland, Iran, Iraq, Mexico, South Africa, Puerto Rico, and the Dominican Republic. These reports suggest that these variants were not only reported in this study (India) but also at times from other countries; however, due to several private mutations, no WHO name was assigned to these variants.

Nucleotide sequences (n = 104) were compared against the 15.6 million genomes in the SARS-CoV-2 metadata and sequences from the GISAID-EpiCoV database to find related genomes. Out of the 14 Alpha variants uploaded, a total of 921 unique related genomes were found. The closest related genomes were from the B.1.1.7 and B.1 lineages. The majority of them were submitted from Spain, Slovakia, Finland, France, Norway, Denmark, the USA, Sweden, Germany, and the United Kingdom between 23 January 2021 and 26 May 2021. Phylogenetic analysis showed that the sequences were at zero distance from the query sequences. Only one similar hit was found for each of our sequences “S505054”, “S35599”, and “S43905” from India (collected from 12 February 2021 to 28 May 2021); for the other 11 sequences, there were no close matches found from India. The collection locations of the related genomes and the number of sequences reported are shown in the map below in Appendix A.

The present study observed the Alpha variant as the third (13.4%) predominant variant, which is in concordance with the national prevalence (14.2%) during this period (November 2020 to March 2021). However, it was underreported in Andhra Pradesh (9.0%). This is partially explained by the low sample size (344) and the patient population. In the present study, B.1.1.7 (n = 8) was predominantly reported from other parts of Andhra Pradesh and the country, whereas the sub-lineage B.1.1.7.4 (n = 1) was reported only from Andhra Pradesh, and B.1.1.7.8 (n = 5) was the second predominant lineage served in this study and was not reported by other centers in the country (Figure 5).

A Delta variant similarity search showed a total of 2880 unique related genomes across the globe. The closest related genomes were from B.1.617.2, B.1, AY.122, and 17 other lineages. They were predominantly reported in ascending order from Canada, Japan, France, Sweden, South Korea, Denmark, Germany, India, the United Kingdom, and the USA between 1 May 2021 and 28 December 2021. They were between a distance of 0 and 2 from the query sequences. Around 250 similar sequence matches were found from India, and 46 matches (44 matches for our sample ID “S1534704” and one match for each “S1508528” and “S43502”) were submitted from the state of Andhra Pradesh. These 46 sequences belonged to B.1.617, AY.112, and AY.92 lineages collected between August and December 2021. The collection locations of the related genomes and the number of sequences reported are shown in the map below in Appendix A. In the present study, B.1.617.2.4 (42.1%) was the predominant observed lineage, followed by B.1.617.2 (27.7%), and others (B.1.617.2.44, B.1.617.2.35, B.1.617.2.27, and B.1.617.2.99.2) in decreasing order (Figure 5). In contrast, the reported prevalence of B.1.617.2.4 was very low (0.15%) in the state of Andhra Pradesh and in the country as well (0.25%). This could be due to the clustering of the study samples during the second wave, where the Delta variant was predominant and the sample size was low. However, the last two varieties reported in this study were not reported by the state, and B.1.617.2.35 was not reported by the country. This could be attributable to the fast evolution of SARS-CoV-2, particularly the Delta variant, during the second wave and the relatively low percentage of sample sequences during this time (March–July 2021 = 0.3% from India) (source: https://covid19.who.int/WHO-COVID-19-global-data.csv (accessed on 3 July 2023)) [17]. 

The B.1.617.1 (Kappa) lineage was detected in 8 (7.69%) of 104 samples during March and April 2021 and found in 538 matched genomes in the database. The closest related genomes were from the B.1.617.1, Unassigned, and B.1 lineages. They were collected in India between 10 March 2021 and 17 August 2021. They were at a distance of “0” from the query sequences. Notably, more than 60% of global sequences were reported from India alone; however, the overall Kappa share was less (1.62%; India). The study found three similar matches from this locality, and the reported percentage (0.77%) was less than the country average (Figure 5). The collection locations of the related genomes and the number of sequences reported are shown in the map below in Appendix A.

The study observed that BA.2, BA.2.10.1, BA.2.56, and B.1.16 lineages under the Omicron variant had close to 2500 unique related genome matches in the EpiCoV database. The majority of them were submitted from the United Kingdom, India, the USA, South Korea, Reunion, France, Australia, Canada, Germany, Japan, Canada, Australia, Spain, Brazil, Belgium, Thailand, and Denmark between December 2021 and July 2022. They were at a distance of 0 from the query sequences. Around 250 exact matches were reported from India; however, from Andhra Pradesh, there were no exact matches found, but at distance “1”, six sequences were observed (one match (BA.2.10) for sample S1541972, three matches (BA.2.10.1) for sample S1565444, and two matches (2.10) for sample S43592). More than 30 lineages and sub-lineages reported by the state were not found in this study, and two lineages (BA.2.56 and B.1.16) reported in this study were not reported by the state (Figure 5). The collection locations of the related genomes and the number of sequences reported are shown in the map below in Appendix A.

Among the other B.1.617 (n = 3), B.1.36 (n = 3), B.1 (n = 2), B.1.551 (n = 2), B.1.1 (n = 1), and XU-recombinant (n = 2) lineages observed in the present study, all were also reported by Andhra Pradesh except the XU variant (Figure 5). The B.1.617 neighbor L452R, E484Q, D614G, and P681R mutations in the RBD of spike protein These mutations, particularly L452R, contribute to a decreased (2.5–4.7 fold) susceptibility of the virus to the neutralization by antibodies (convalescent sera, monoclonal antibodies, vaccination) and more efficiently entering the host cells [18,19]. In the present study, the majority of Delta and Kappa variants and B.1.1 lineages also demonstrated these mutations (Figure 6). The collection locations of the related genomes and the number of sequences reported are shown in the map below in Appendix A. The findings of this study (above five paragraphs) are based on metadata associated with 15.6 million sequences available on GISAID up to 28 May 2023 and accessible at https://www.epicov.org/epi3/frontend#414f51 (accessed on 3 July 2023) [20,21].

During early 2020, a study from Andhra Pradesh [22] reported SARS-CoV-2 variants majorly clustered (94%) under the clades 20A, 20B, and 20C, while others fell under the I/A3i clade (6%). A similar study reported from India showed that soon after these clades, the Delta variant (42.15%) was the most predominant lineage, followed by Kappa (10.06%) and Alpha (7.22%) [23], until September 2021. Soon after two months, an Omicron upsurge with two variants, BA.1 (December 2021–January 2022) and BA.2 (January–May 2022), was predominant in India [24]. The “NIH OPA iSearch COVID-19 Portfolio” database search with key words “covid AND sars” revealed 337,699 publications globally between 1 January 2020, and 10 July 2023. Out of these, 4396 were from India and 67 from the state of Andhra Pradesh (with the key words “covid AND sars cov2 AND andhra pradesh”), and none of these studies exclusively reported the data from this state [25].

There were 69 types of amino acid substitutions reported (excluding private mutations) throughout the spike gene (Figure 2). Aspartic acid in position 614 was replaced by glycine (D614G), which was noted to be increasing in frequency since April 2020 and was also observed in all 104 samples in the present study. This mutation was thought to be a positive selection, which helps in the virulence, infectivity, and transmissibility [26,27] of the virus. Another N-terminal domain (NTD) of the spike protein mutation G142D was the next common mutation (n = 78) observed during the study, which was consistent with nearly all samples of Delta, Kappa, and Omicron variants. Strikingly, this mutation was reported to be associated with low cycle threshold values (Ct) along with the T95I mutation in infected samples (higher viral load), and frequent back mutations were also reported. The T95I mutation was observed predominantly in Delta variants (15/38), followed by Kappa (3/8) and Omicron (1/31). These two mutations play a major role in the conversion of a strand to a helix structure around aa159–167 and aa183–190, along with other alterations that affect the antibody binding epitope (7L2E, 4A48, and 7C2L), resulting in decreased antibody affinity for this region [28].

The following mutations (H146Y, A263P, N481K, V511I, T604N, and K1245R) observed in the present study were not reported by the state of Andhra Pradesh. A spike mutation H146Y observed in one of the Delta variant sequences may play a role in decreased susceptibility to neutralization [29]. Both national and global reported percentages of the other five mutations were less than 0.01% (Appendix A). The first four mutations were observed in different lineages of the Delta variant, and the last two were observed in the Kappa and Omicron variants, respectively (Figure 6 and Appendix A). The percentage and ratio of each mutation (estimated by dividing the percentage of a particular mutation observed in this study with the percentage of the same mutation reported globally) were tabulated to show how many times more or less a given mutation was observed in this study as compared to local, national, and global data. In the present study, a few mutations (K77T, V382L, S929I, R158G, N481K, V511I, T604N, and K1245R) were observed to be several (40–250) fold higher than the national and global data, whereas the other mutations (V213G, E484Q, E154K, Q1071H, and H1101D) reported in our study and national data were 40- to 400-fold higher than global prevalence. The majority of these peculiar mutations were observed in B.1.617.2, B.1.617.2.35, B.1.617.1, B.1.617.2.4, B.1.617.2.44, and B.1.617.1 lineages (Figure 6 and Appendix A). This is largely attributable to the relatively higher share of Delta and Kappa variants during the second wave, and during this period more samples were sequenced as a part of close monitoring of emerging VOCs and lineages.

SARS-CoV-2 spike protein residues from 300 to 530 form the receptor binding domine (RBD); within this sequence, the receptor-binding motif (RBM) consists of 424 to 494 residues. Leucine (L445), phenylalanine (F486), glutamine (Q493), serine (S494), and asparagine (N501) are the five critical residues in the RBM of SARS-CoV-2 that interact directly with the host receptor angiotensin-converting enzyme 2 (ACE2) [30]. Nearly all Alpha and Omicron lineages had the N501Y substitution; Q493R was observed only in the Omicron lineages; and other mutations (L445, F486 and S494) were not observed in the present study. However, both the diversity and rate of mutation were higher in the RBD region, especially in all lineages of the Omicron variant. In the S2 subunit of the SARS-CoV-2 spike protein, H655Y, N679K, P681H, N764K, and D796Y (fusion peptide region), Q954H and N969K were commonly observed mutations, and nearly all Omicron variants exhibited these mutations (Figure 6). Many of these mutations (N679K, P681H, N764K, and D796Y) found in the Omicron variant are located near the furin cleavage site and are believed to play a role in spike protein cleavage, increase virulence, prevent protease recognition, decrease syncytia formation, divert cell entry through the cell surface to the endocytic pathway, antibody evasion, and viral transmission of the strain [31,32,33].

To elucidate the interaction between the SARS-CoV-2 spike and human ACE2, we determined the structure using homology modeling. Some interactions between SARS-CoV-2 RBD and human ACE2 have previously been identified; however, the actual residues involved in the interactions remain unclear [34]. Several studies have revealed major interactions between SARS-CoV-2 RBD and ACE2 with X-ray crystallography [35]. But the actual residual interactions are changing from time to time based on mutations in the spike gene during the pandemic. The mutation rate in the SARS-CoV-2 spike gene differs from Alpha to Omicron and other emerging variants. Our study mainly focused on the structural integrity at the residual level and amino acid variations at the RBD region for each variant type. Trimer structures were built for spike proteins of 66 unique sequences, interactions between the residues were determined (Appendix A), and mutations were labeled on the built structures of representative variants (Appendix A). 

Computational (protein–protein docking) studies revealed that the mean interactions between the A, B, and C chains of the spike protein trimer structures of the four variants reported in this study progressively decreased from the Alpha variant (mean = 213.5) to the Omicron (mean = 204.5) (Appendix A). We found a statistically significant difference in average bonded interactions between the A, B, and C chains in the spike protein trimer structure (F (3, 87) = 8.606, *p* < 0.001). There was no significant difference observed between the means of non-bonding interactions (F (3, 87) = 0.4182, *p* < 0.7404) (Appendix A). Similarly, interactions between the spike protein (trimer) and the hACE2 receptor were also analyzed. It was observed that both the number of bonded and non-bonded interactions between the spike and the receptor decreased from the Alpha variant (mean = 24.07 and 254.5) to Kappa (mean = 14.0 and 205.0) and increased to 27.12 and 311.12, respectively, in the Omicron variant. The observed difference was statistically significant (bonded (F (3, 87) = 9.311, *p* < 0.0001)) and non-bonded (F (3, 87) = 10.69, *p* < 0.0001)) (Appendix A and Appendix A).

Among the four VOCs, the Delta variant was responsible for severe disease and mortality in all age groups. On the other hand, the Omicron variant displayed more infectivity and less mortality as compared to the Delta variant. Compared to other variants, Omicron showed more interactions, both bonded and non-bonded. The increase in non-bonded interactions could be due to the formation of two additional salt bridges resulting from the exchange of glutamine 493 to arginine (Q493R). This substitution led to a strong increase in linear electrostatic interaction energy and an increase in the probability of RBD-ACE2 interaction, thereby increasing infectivity [36]. It was postulated that mutations in K417N, Q493R, N501Y, and Y505H residues cumulatively contributed to direct changes in interaction with the receptor; further, these mutations, along with other mutations (N440K, T478K, and Q498R) within the RBD region, make it more electropositive than other variants. It is well established that the higher the positive charge of spike protein, the better the binding to the negatively charged glycocalyx (heparin sulfate) [37,38]. In conclusion, Omicron started to dominate the pandemic at the end of 2021, with patient numbers rising steeply in affected areas. Computational analysis of the spike-ACE2 complex shows clear differences between the Omicron variant (more interactions) and the Delta variant, explaining its strong binding to receptors and immune escape. 

### Effect of Vaccination on Mutations

After successful clinical trials, the first COVID-19 vaccine (Pfizer BioNTech, New York, USA) was introduced on 8 December 2023 [39]. A month later, on 16 January 2021, the Indian government started a vaccine campaign with two vaccines, Covishield (AstraZeneca-Oxford University-Serum Institute of India) and Covaxin (Bharat Biotech, Hyderabad, India), and achieved 6 million beneficiaries in just 24 days [11]. Due to the short supply of the above vaccines and the limited availability of other vaccines (Pfizer BioNTech, New York, NY, USA and Moderna, MA, USA), during the first few months these vaccines were restricted to only front-line and health-care workers. Out of 104 samples subjected to sequencing, 57 were from the second wave (Alpha-14, Delta-27, Kappa-8, and others-8). The data show that the second wave was predominantly caused by the Delta variant. The later vaccines were known to be more effective on the Delta variant but were not available in India during that time [40]. As the sample size of this study was insufficient to draw any conclusions on the effect of vaccines on the occurrence of new variants and vice versa, further studies are warranted. 

Several studies have revealed that some crucial mutations, like N501Y in spike protein within RBD, are sufficient for higher affinity to human ACE2 [41,42,43], enhance resistance to neutralizing antibodies, and increase virulence capacity [44]. Our study revealed that mutation rates from Alpha to Omicron variants increased dramatically (Appendix A) at the NTD and RBD regions as compared to other portions of the spike protein. This evidence suggests that eventually, mutations in the NTD and RBD regions helped the virus evolve into a dominant variant in circulation among host populations.

Two main components in the S1 domain are RBD and N, the terminal domain (NTD). The following mutations were observed in the NTD: L5F(1), Q14H(1), L18F(1), T19R(37), T19I(32), A27S(33), H49Y(1), A67V(2), K77T(13), T95I(22), G142D(78), Y145D(2), H146Y(2), E154K(8), R158G(36), V213G(34), A222V(3), A262S(1), and A263P(1). In the Delta variants (34/38) threonine in the 19th position was replaced by arginine (T19R); after that, it was replaced by isoleucine (T19I) in the Omicron variants (30/31). A27S was reported [45] to be a unique mutation seen in Omicron sub-variant BA. 2. Consistent with this observation, in the present study, 30 out of 31 Omicron BA. 2 variants had this mutation, which was not found in other variants. The NTD plays a major role in the point of recognition for vaccines, attachment, and the immune response. Higher-numbered NTD mutations, such as A67V, V70F, G142D, del 211, and ins214EPE, result in structural changes and contribute to antibody evasion. Therefore, any changes in amino acid sequence in this region are expected to have deleterious effects on viral immunity, leading to evasion of immunity and a decreased effect of vaccine-induced immunity for the newer variants [33,46,47,48].

In the present study, a total of 900 SARS CoV-2 Real-time RT-PCR positive samples (archived) were intended to be subject to spike gene sequencing. During pre-sequencing steps 796 samples were excluded due to various reasons (high (>25) Ct value = 398, failure of one or more targets in the end-point PCR = 398). Since the Sanger sequencing method can process long fragments (900–1000 bp) of Amplicons as compared to some next generation sequencing (NGS) methods, to reduce the number of PCR reactions in the present study, the primers were designed to target and amplify the large size spike gene fragments from 941 to 1282 bp. We believed, due to the poor integrity of RNA in stored samples, that despite the rRT-PCR positive end point, PCR failed to amplify and produce sharp bands (in agarose gel) in all six targets. Therefore, 104 samples were subjected to downstream processes. Though the sample size is smaller, we reported several unique lineages and sub-lineages which were not reported in this region, and this is the first research study on mutation analysis in this region. However, more studies are warranted to support the findings of this study.

## 4. Conclusions

The present study was conducted to investigate the occurrence of various SARS-CoV-2 variants in this part of the state during the first, second, and third waves of the COVID-19 pandemic. In the early pandemic, wild type and Alpha variants were detected, and the Alpha variant continued to be present till the mid-second wave (May 2021). Later, it was replaced predominantly by the Delta variant, followed by Kappa. Unlike the second wave, during the third wave, the BA.2 lineage of Omicron was reported predominantly (84%), followed by BA.2.10.1 (9.6%), BA.1.1.16 (3.2%), and BA.2.56 (3.2%). The positivity trend (disease outbreaks) was in line with the national data, and there were four clear waves (1–4) observed. The first wave was broad (10 months), with daily positivity reported at around 15–25% during peaks. As compared to the first wave, the second wave was less broad (4–5 months) with 20–30% daily positive cases. Unlike the first two waves, the third wave was short (2–3 months), but the positivity reached up to 50%. Though the number of samples processed has been less than 50 samples per day since March 2022, positive cases were detected during June–September 2022, which could be considered the fourth wave. MD studies revealed the mutations were predominantly accumulated on the NTD and RBD of spike proteins. Similarly, the Omicron variant showed more interactions between the spike and ACE2, explaining its high infectivity and positivity. As per the GISAID database, out of the 104 samples analyzed in this study, similar sequence matches from India were found for 15 samples, and for the remaining 89 samples, there were no identical sequences reported from the country. The study observed a few peculiar spike gene substitutions (H146Y, A263P, N481K, V511I, T604N, and K1245R), previously not reported from this state. Therefore, the data generated by this study will be a new addition to India’s SARS-CoV-2 database.

## Figures and Tables

**Figure 1 viruses-15-01656-f001:**
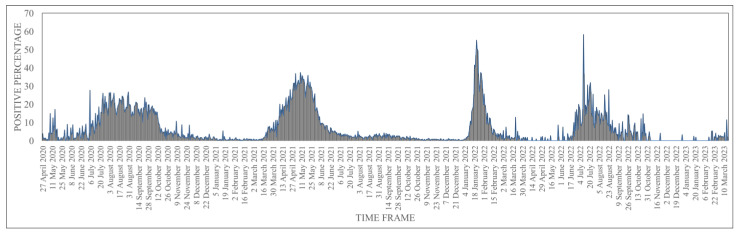
Trends of COVID-19 peaks (positive percentage) observed in the Rayalaseema districts of Andhra Pradesh state based on the laboratory data The time frame (April 2020 to March 2023) is shown on the “X” axis, and the number of positive cases reported per day is shown as percentages on the “Y” axis.

**Figure 2 viruses-15-01656-f002:**
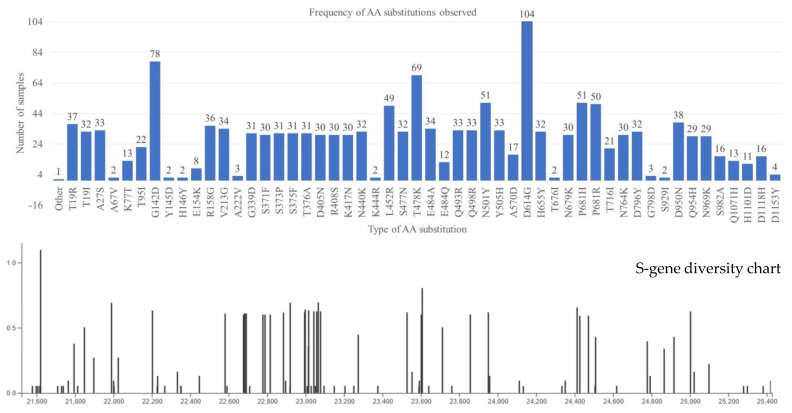
Amino acid substitutions and their frequency are represented as blue bars (**up**). The height of the bar corresponds to the number of samples exhibiting the mutation. Some mutations were observed in only one sample each, these mutations are presented as the others and include; L5F, Q14H, L18F, H49Y, A262S, A263P, V382L, L452M, N481K, G496S, V511I, T547K, T604N, Q675H, N856K, and L981F. The genomic diversity chart (**down**) displaying the nucleotide substitutions for 104 isolates. The scales below the chart (21,600–25,400) represent the corresponding nucleic acid positions, respectively, of the spike gene.

**Figure 3 viruses-15-01656-f003:**
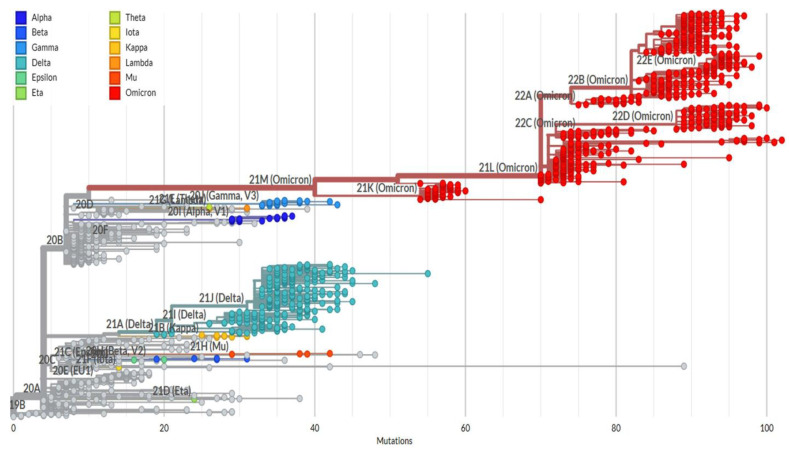
A rectangular phylogeny tree (without roots) was constructed using a local-browser-dependent Nextclade program (updated: 14 December 2022). Sequences (complete spike gene) from the present study were placed (colored dots) on a reference tree (Wuhan-Hu-1/2019 (MN908947)). The Nextstrain and WHO clade names are shown on the respective colored dots.

**Figure 4 viruses-15-01656-f004:**
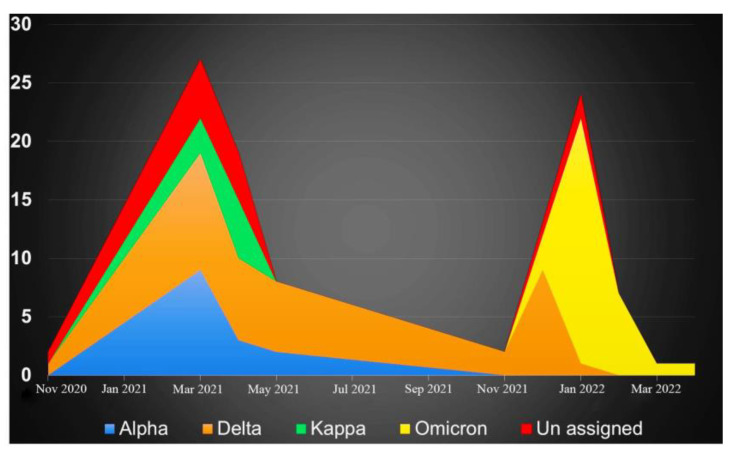
The distribution of the occurrence of each Alpha (n = 14; blue), Delta (n = 38; orange), Kappa (n = 8; green), Omicron (n = 31; yellow), and others (n = 13; red) over a period of time is shown on the “Y” axis; their occurrence in the time frame is indicated on the “X” axis.

**Figure 5 viruses-15-01656-f005:**
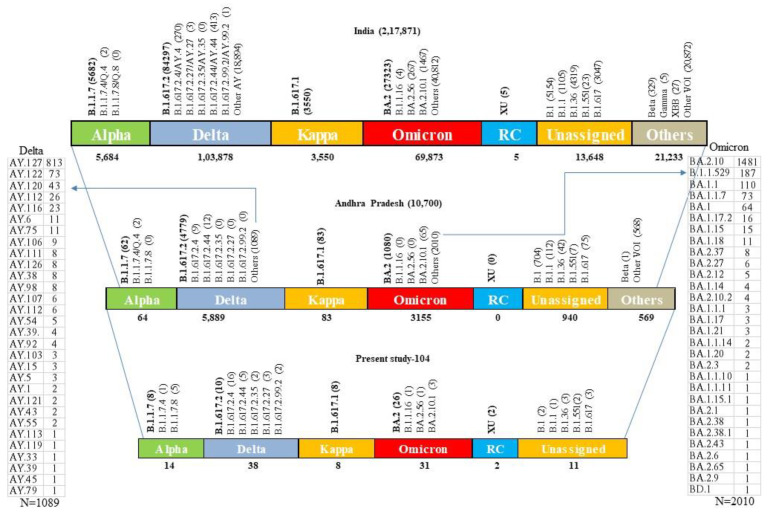
Lineages and sub-lineages reported in the present study are compared with the local (Andhra Pradesh state) and national (India) data. “Unassigned” are sequences that could not be assigned a lineage by Pangolin. Lineages highlighted in the bold text/numbers are major lineages reported from this study and the Country as well. Additional lineages of Delta and Omicron variants reported from the state are shown on the left- and right-side columns, respectively.

**Figure 6 viruses-15-01656-f006:**
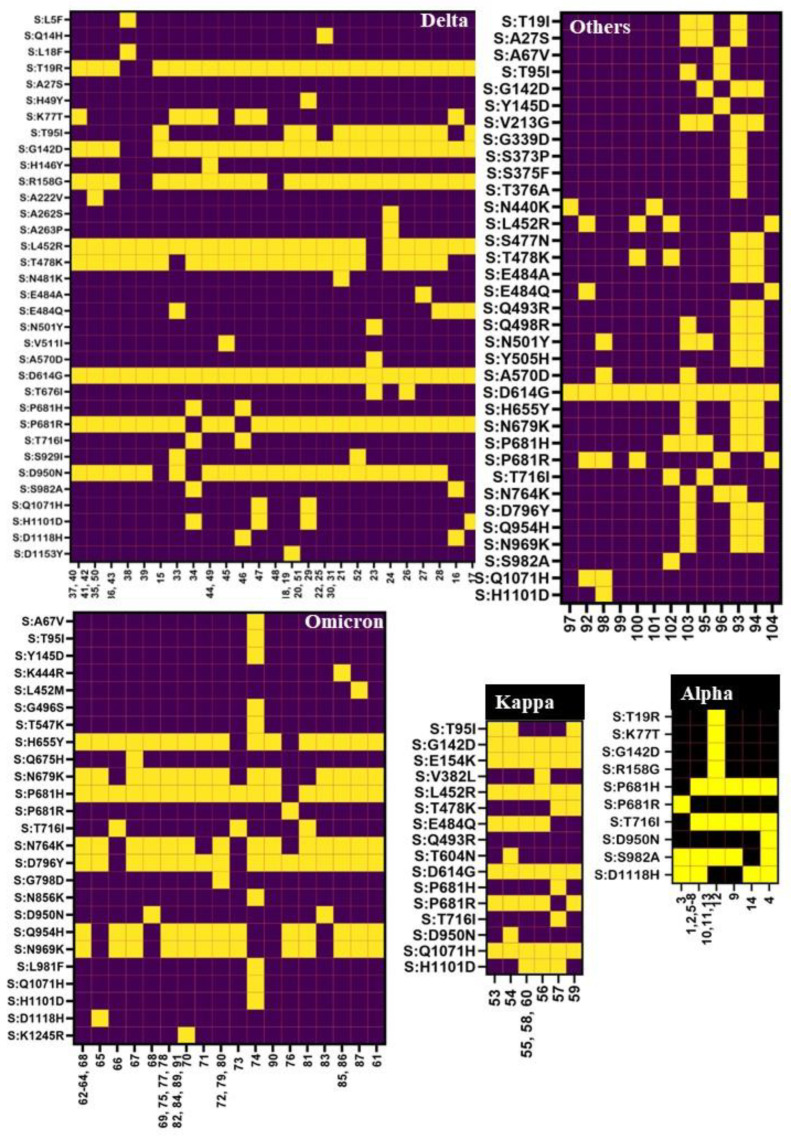
Maps representing the peculiar amino acid substitutions observed in Alpha (14), Delta (38), Kappa (8), Omicron (31), and other (13) variants reported in this study. Each row on the “Y” axis represents a specific amino acid change, and squares filled with yellow color indicate a mutation. Sample numbers (corresponding nucleotide sequence GenBank accession numbers and lineage names are provided in the Appendix A) are represented on the “X” axis. Some substitutions were observed among all lineages of the varietal but are not shown in the map. These substitutions are as follows: Alpha-S:N501Y, S:A570D, and S:D614G; Omicron-S:T19I, S:A27S, G142D, S:V213G, S:G339D, S:S371F, S:S373P, S:S375F, S:T376A, S:D405N, S:R408S, S:K417N, S:N440K, S:S477N, S:T478K, S:E484A, S:Q493R, S:Q498R, S:N501Y505H, and D614G.

**Table 1 viruses-15-01656-t001:** Flanking regions of primers used for pre-sequence amplification and their amplicon sizes.

Target	Primers		Primer Binding		Target Size	Overlap
			Start	End		
T1	FP *	ACAAATCCAATTCAGTTGTCTTCCTATTC	21,357	22,326	969	364
	RP	CACCAGCTGTCCAACCTGAAGA				
T2	FP *	CAATTTTGTAATGATCCATTTTTGGGTGT	21,962	22,903	941	386
	RP	ACCACCAACCTTAGAATCAAGATTGT				
T3	FP *	AGAGTCCAACCAACAGAATCTATTGT	22,517	23,522	1005	399
	RP	CAGCCCCTATTAAACAGCCTGC				
T4	FP *	CCAGCAACTGTTTGTGGACCTA	23,123	24,126	1003	337
	RP	CATTTCATCTGTGAGCAAAGGTGG				
T5	FP *	GTGGTGATTCAACTGAATGCAGC	23,789	24,789	1000	398
	RP	GTGAAGTTCTTTTCTTGTGCAGGG				
T6	FP *	GCACTTGGAAAACTTCAAGATGTGG	24,391	25,673	1282	390
	RP	AGGTGTGAGTAAACTGTTACAAACAAC				

* Sequencing primers; reference SARS-CoV-2 S-gene range from 21,563 to 25,384 bases.

## Data Availability

Both complete and partial S-gene sequences are deposited in NCBI GenBank. Accession numbers for the complete S-gene are “ON644350-ON644384, ON651690-ON651724, and ON668127-ON668160”. Accession numbers for partial S-genes are “ON644612-ON644868, ON651730-ON652014, ON652017-ON652303, and ON680851-ON680920”.

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
