# Peer review of "Mutation Analysis of SARS-CoV-2 Variants Isolated from Symptomatic Cases from Andhra Pradesh, India"

_viruses, 2023, doi:10.3390/v15081656_

Round 1
Reviewer 1 Report (Previous Reviewer 2)
The authors analyzed SARS-CoV-2 variability in the State Andhra Pradesh of India. For this, they sequenced the S gene to determine the variants and analyzed the mutations present in 104 samples. This study is a resubmission: the authors intended to address the previous concerns of the reviewers. However, not all concerns were addressed satisfactorily, so the manuscript need further revision.
1. From March 2020 to April 2022 (period of this study), a total of 217.862 sequence from India are available in GISAID. From these, 10.700 are from Andhra Pradesh: 62 Alpha, 5867 Delta and 3079 Omicron. The authors should compare the sub-Pango lineages inside each of these VOC and compared them with the one they found in their study. This would be some kind of validation of their study.
2. In addition, the frequency of Alpha VOC is higher in their study, compared to the one found in GISAID for the locality from November 2020 to March 2021 (30/344 seq), period of time when this variant peaked. The authors should discuss this.
3. Some isolates could not be assigned to a particular variant. This is somehow surprising, particularly at the end of 2021-beginning of 2022. The authors should have a look of the amino acid mutations characteristic of each variant to try to infer at least the major variant lineage of them.
4. In the Results and Discussion, too many details are provided that could be deleted: for example, there is no need to mention if the amplicons were purified from gel or not.
5. The historic description of the pandemic in the World, in Results, should also be deleted.
6. The legends do no correspond to the figures and need correction.
7. Figure 1 is not clear and illegible. Why there is such differences between cases and number of samples tested? This is another limitation of this study that should be mentioned.
8. Figure 2 (the legend is in Figure 4) is also illegible and not very meaningful.
9. Figure 5-9 do not bring any relevant information. Many of the mutations described have already been described for each sub-lineage of the VOCs Delta and Omicron. The authors should focus in peculiar mutations, if any, and not in the common mutations of each VOC/sub-lineage.
10. The sub-lineages of Delta are not described, just the number mentioned.
11. Lines 435-437. Please edit for syntaxes. Not all the mutation indeed, some of them.
12. The authors do not cite previous studies on variant surveillance in India. This should be included, with particular comparison with the region.
13. In general, the authors should focus in bringing to the audience clear messages from their study: utility of the method developed for variant study, the description of the sub-lineages observed in the regional context, and describe better the modeling results, to provide some message from this analysis. Without this focus, the manuscript is still lacking of interest for the audience.
Author Response
Dear Sir/Madam,
please find the authors response attached.

Reviewer 2 Report (New Reviewer)
This is a survey of mutational variation of sars-cov-2 in India. 104 samples were sequenced for a single gene. The analyses generally seem valid, but it is a very small sample of sequences relative to most modern genomic studies on sars-cov-2.
The English is okay; things can always be improved...
Author Response
Dear Sir/Madam,
I would like to convey our sincere thanks for your time and the valuable suggestions to improve the manuscript and guiding us to bring out kay findings for the interest of readers/scientific community.
Reviewer 3 Report (New Reviewer)
The paper presents the story of the investigation of the occurrence of various SARS-CoV-2 variants in Andhra Pradesh, India during the three waves of the pandemic. The research is based on samples taken by the authors' lab which was the first COVID-19 testing lab in the state. The paper is written clearly, and illustrated with charts showing the results that support the study. I have just a few comments for the authors:
Major questions and comments:
(1) The description of the structure of SARS-CoV-2 is rather poor, I suggest adding at least information about research aimed to determine its 3D structure (including those on experimental and computational sides), sample works to cite: Lan et al., 2020 (doi: 10.1038/s41586-020-2180-5), Gumna et al., 2022 (doi: 10.3390/ijms23179630), Rangan et al, 2020 (doi: 10.1261/rna.076141.120), Wang et al, 2020 (doi: 10.3389/fcimb.2020.587269).
(2) Please, add references to all software tools and data sources that were used in the project, for example:
- ClustalX (Larkin et al, 2006; doi: 10.1093/bioinformatics/btm404)
- Modeller v10.4 (at least a webpage)
- Protein Data Bank (any paper of your choice, there are many about PDB)
- ClusPro (Kozakov et al., 2017; doi: 10.1038/nprot.2016.169)
etc...
(3) Figure 1: please, reverse the colors, the drawing will be more readable if the dark graph is on a white background.
Minor comments:
(1) In the graphical abstract:
- remove the header "Conclusions" (it is not necessary)
- "104 Samples selected & Sequenced for S gene." -> 104 samples sequenced for the S gene.
- "results in structural changes" -> result in structural changes
(2) Remove multiple spaces that occur between the words, for example:
" poses a risk"
"shares 79%"
"bars). Number"
"energy and increase"
"leading to decreased"
etc.
(3) The spelling of the virus name should be standardized. Currently, authors use several versions - SARS-CoV-2, SARS CoV2, SARS-CoV2. They should decide on one and use it consistently throughout the work.
Also, unify the spelling - S-gene or S gene.
(4) "genome and submitted to NCBI GenBank" - please, specify the accession code(s)
(5) Some other suggested corrections are:
- "positive per cent" -> positive percentage
- " (non-bonded ((F (3, 87) =10.69, p < 0.0001)). supplementary figure 9c-d and supplementary table 1." -> (non-bonded ((F (3, 87) =10.69, p < 0.0001)) (Supplementary Figure 9c-d and Supplementary Table 1).
Some corrections are needed (as indicated in the previous section).
Author Response
Dear Sir/Madam,
I would like to convey our sincere thanks for your time and the valuable suggestions to improve the manuscript and guiding us to bring out kay findings for the interest of readers/scientific community. As you suggested we tried to do our best to improvise the manuscript and submitted the revised manuscript. Please let us know if any more revisions are required in the manuscript.

Round 2
Reviewer 1 Report (Previous Reviewer 2)
The authors addressed adequately the concerns.
There are still minor synthax errors.
For example in the Abstract:
This study was aimed to detect various SARS- 34 CoV-2 variants circulating in the state Andhra Pradesh, India. During 2020-22 close to 1.8 million 35 samples were tested, complete S-gene was sequenced (Sangers method) from 104 samples, based- 36 on the S-gene nucleotide sequence mutations were analysed.
After were tested, strong poncuaitons is needed (.)
After 104 samples also. The next sentence is not clear.
There are others English editing needed.
Author Response
Dear Sir/Madam,
Thank you for your quick review and suggestions. We have corrected all syntax and grammar errors to the best of our ability. The revised manuscript is attached for your kind review and suggestions.
This manuscript is a resubmission of an earlier submission. The following is a list of the peer review reports and author responses from that submission.
Round 1
Reviewer 1 Report
Overall, the paper is well-written and interesting, however, it needs improvement.
1- Graphical presentation of the data are poor, please revise the figures (particularly fig 1a and 1b) and make them easier to read.
2- The result and Discussion section is written poorly, need revision, the data need to be compared with the different SARS-COV-2 variants recorded within the same period of time.
3- This article did not include the effect of vaccination program on appearing of different SARS-COV-2 variants and/ or controlling the spread of virus. Please add this to your discussion section.
Reviewer 2 Report
The authors analyzed SARS-CoV-2 variability in the State Andhra Pradesh of India. For this, they sequenced the S gene to determine the variants and analyzed the mutations present in 104 samples. This study has several limitations that hampered its acceptance.
Major comments:
1. The number of samples is low and only the S gene region was sequenced, reducing the effectiveness of the analysis.
2. The information related to the mutations found is somehow descriptive but without a clear message. An example of this are the Conclusions.
3. The authors did not even perform a comparison with other studies from India. In January 22, 314,731 sequences of Indian isolates have been deposited in GISAID.
4. The rate of positivity shown in Figure 1b was near 0 several times during the pandemic, which is surprising, while many cases were tested (Figure 1a). This is not discussed and a problem with the real-time diagnostic test is suspected.
Minor comments
5. The title should not say India but specify the state where the study was performed.
6. Table 4 should be more suitable as a Supplemental table.